# Reflexive Gaze Shifts and Fear Recognition Deficits in Children with Callous-Unemotional Traits and Impulsivity/Conduct Problems

**DOI:** 10.3390/brainsci11101342

**Published:** 2021-10-13

**Authors:** Luna C. Muñoz Centifanti, Timothy R. Stickle, Jamila Thomas, Amanda Falcón, Nicholas D. Thomson, Matthias Gamer

**Affiliations:** 1Department of Primary Care and Mental Health, University of Liverpool, Liverpool L69 3BX, UK; 2Department of Psychological Science, University of Vermont, Burlington, VT 05405, USA; jamila.thomas@gmail.com (J.T.); amanda.falcon@uvm.edu (A.F.); 3Division of Acute Care Surgical Services, Department of Surgery, Virginia Commonwealth University, Richmond, VA 23284, USA; nicholas.thomson@vcuhealth.org; 4Department of Psychology, University of Würzburg, 97070 Würzburg, Germany

**Keywords:** callous-unemotional traits, eye-tracking, emotions, conduct problems, emotion recognition

## Abstract

The ability to efficiently recognize the emotions on others’ faces is something that most of us take for granted. Children with callous-unemotional (CU) traits and impulsivity/conduct problems (ICP), such as attention-deficit hyperactivity disorder, have been previously described as being “fear blind”. This is also associated with looking less at the eye regions of fearful faces, which are highly diagnostic. Previous attempts to intervene into emotion recognition strategies have not had lasting effects on participants’ fear recognition abilities. Here we present both (a) additional evidence that there is a two-part causal chain, from personality traits to face recognition strategies using the eyes, then from strategies to rates of recognizing fear in others; and (b) a pilot intervention that had persistent effects for weeks after the end of instruction. Further, the intervention led to more change in those with the highest CU traits. This both clarifies the specific mechanisms linking personality to emotion recognition and shows that the process is fundamentally malleable. It is possible that such training could promote empathy and reduce the rates of antisocial behavior in specific populations in the future.

## 1. Introduction

Quickly assessing or “reading” emotions in others facilitates social communication and empathic responding. Children and adults who lack empathy appear to have trouble accurately reading and labelling fearful expressions when looking at other people’s faces. These individuals are described as displaying callous-unemotional (CU) traits, which have been added to the Diagnostic and Statistical Manual of Mental Disorders [1] as “Limited Prosocial Emotions,” to specify a subset of children with conduct disorder. CU traits include lack of caring about what typically developing people care about, such as performance at school and/or work, as well as caring about the effect their behavior has on others. In contrast, people with these traits show a callous disregard of others in gaining favors for themselves, as well as a lack of emotional depth. Conduct problems and antisocial behavior that is of an aggressive nature often characterizes those who are callous and unemotional, because these individuals do not know (or care) when they cause distress in their victims. Thus, difficulties reading fearful facial expressions may, in part, explain this lack of cognitive understanding of other people’s emotions and, therefore, also aggressiveness toward others. It is important to understand what contributes to inaccurate reading of emotions so we can design effective interventions to reduce or correct this deficit. Toward this end, we have conducted two studies within two separate populations of youths showing conduct problems and antisocial behavior to understand whether youths with CU traits attend to the most salient and diagnostic information about fear (i.e., eyes) and whether directing them to look at the eyes (using passive and active methods) can improve their recognition of fear. 

Several empirical studies have found CU traits to be related to fear recognition deficits, such that children with these traits have been described as being “fear blind” [2] or generally unable to read fearful expressions. Those with CU (and psychopathic-like) traits experience difficulty recognizing fearful emotional expressions (and sometimes sadness) in others, but deficits in reading and understanding other emotions are not consistently related to CU traits [2,3,4,5,6]. These emotion reading deficits relate to the general empathy problems shown in those with CU traits. A key characteristic of CU traits in children and adolescents is believed to be deficient empathy, e.g., [7]. Although some studies suggest attention towards eyes may underlie these emotional processing deficits [8], there is a lack of research into how (and how effectively) facial cues are used by people with CU traits for identifying fearful expressions. 

### 1.1. Fearful Expressions Are Determined by Looking at Eye Regions

People with CU traits may show deficits in reading fearful expressions in faces, simply because they have reduced shifting of attention (via eye gaze) towards the eye region of others [2]. The eye regions of faces contain the most salient cues for fear recognition—there is greater proportion of the visibility of the sclera of the eyes which acts as an aid [9]. Using eye-tracking methodology, impaired recognition of fear was associated with reduced sampling of the eye region of other people [10]. Recognition of fear and other facial cues is achieved via a distributed neural system, e.g., [11]. For example, as noted by Jhou et al. [12], several related systems are involved including visual (middle/superior and medial temporal gyrus), prefrontal (medial frontal gyrus and middle frontal gyrus), and limbic regions (amygdala, parahippocampal gyrus). The work of these authors also shows core activation in those areas in response to facial stimuli. Additionally, low inhibitory control is, in part, related to the hippocampal system via GABAergic interneurons and rostromedial tegmental nucleus in its role as a “brake” in the midbrain dopamine system [12]. Thus, there are both focal areas of interest and distributed systems that are involved in processing and that may be related to fear blindness. Certainly, if fearful faces are construed as important to learning to inhibit one’s behavior that is leading to others being afraid, then the involvement of these systems in inhibition and punishment reactivity is key to our understanding. Interestingly, the behavioral findings in children with CU traits parallel observations from patients with bilateral amygdala damage, who show reduced gaze orienting towards the eye region of other people, including the sclera of the eyes [13,14] as well as deficits in the recognition of fearful facial expressions [15]. On this basis, it has been speculated that the amygdala may be dysfunctional in individuals with CU traits [16,17]. 

Support for the idea that reduced attention to the eyes underlies the deficits seen in those with CU traits is bolstered by evidence indicating that attentional factors (such as attention to particular areas of the face that convey emotion) are important in modulating the fear deficit established in prior studies [17]. For example, Dadds et al. [2] showed that focusing attention on the eyes aided fear recognition in youth highest in CU traits, i.e., the group in the upper quartile of the sample. In this task, youths were directed to look at the eyes while performing a facial emotion recognition task within a single testing session. Although benefits were demonstrated during the verbal gaze instruction, the benefits in fear recognition were not maintained when the instruction was no longer used. These results suggest that focusing attention to the eyes can ameliorate this emotion processing deficit, but children with CU may not recognize this strategy as useful or critical to the task. Alternatively, they may be unmotivated to use the strategy, or simply fail to remember to use it when not explicitly instructed to. Thus, the verbal instruction may have tasked children with CU traits to use an unfamiliar strategy that they had to voluntarily put into practice, but they may not reflexively know (or indeed care) that this strategy results in better performance when trying to read fearful expressions in other’s faces.

### 1.2. Reflexive Shifts in Eye Gaze as Automatic Attention

Previous experimental paradigms frequently involved presenting facial expressions in the center of a computer screen and allowing for a free exploration of these stimuli for several seconds. Free-gaze paradigms such as this typically begin with fixations in the middle of the face and have a long viewing period to calculate the total time spent looking at the eye region or number of gaze shifts, or saccades, to the eye region. Although such studies can provide evidence on the differential exploration of different informative or diagnostic features of a face (e.g., the mouth region for identifying happiness or the eye region for fear), they do not allow for differentiating between reflexive aspects of gaze orienting and more sustained processes of attentional exploration. It may be that children with CU traits initially look at the mouth, avoiding the most important fear cue—the eyes—and this initial saccade may influence patterns seen in prior studies. Indeed, eye-tracking results can be greatly affected by the first saccade [9], and subsequent viewing behavior may depend on where people first look, making it difficult to know if random, undirected saccades drive the differences described for those low and high on CU traits. Thus, prior studies have not clarified whether children with CU traits might be initially and automatically scanning faces differently.

Therefore, we devised an experimental procedure that includes a systematic manipulation of the initial fixation [18]. In addition to providing information on visual exploration patterns (i.e., sustained attention), this procedure allows for isolating reflexive aspects of visual orienting by examining initial shifts of attention from non-diagnostic (i.e., irrelevant) towards diagnostic (i.e., relevant) features and vice versa. Of importance, enhanced amygdala activation was reported to be associated with gaze shifts from the mouth towards the eye region of fearful faces [9] and amygdala lesions seemed to specifically impair these reflexive shifts of attention [14]. Interestingly, amygdala hyporesponsiveness is thought to be central in the development of CU traits [19,20], so we may see these deficits in reflexive shifts for fear using this paradigm.

### 1.3. Current Study

The aims of the current study were twofold: (1) to investigate whether children with high CU traits show less reflexive attentional orienting towards the eye region of fearful facial expressions in Experiment 1; (2) to extend previous findings on interventions to improve fear recognition abilities in youths with high CU traits [2] in an adjudicated sample of adolescents that included youths with elevated CU traits in Experiment 2. Specifically, we conducted a pilot study to test whether juveniles high in CU traits had sustained improvements in their recognition of fearful faces after being administered an intensive learning intervention that included repeated eye-gaze instruction to specifically “look at the eyes” of face stimuli. As noted above, prior research has used free-gaze paradigms and concluded that children with CU traits spend less time attending to the eyes when freely viewing fearful faces. 

To correct for first-saccade effects in free-gaze paradigms, Gamer and Büchel [9] designed an experimental paradigm that shifted faces up and down on the screen to direct participants to first fixate either on the eye region or mouth region of faces. They then examined whether the first saccade after presentation was directed toward the eyes (when the mouth was fixated) or toward the mouth (when the eyes were fixated). Since presentation time was deliberately chosen to be shorter than the latency of initial saccades (i.e., 150 ms), this paradigm allows for measuring reflexive saccades that are triggered by the stimulus presentation but do not help for visually exploring the stimuli. Based on previous studies, we supposed that youths with CU traits may not automatically attend to the eyes and therefore show deficits in recognizing emotional expressions such as fear that critically depend on extracting information from the eye region of others. If such deficits akin to patients with amygdala damage can be demonstrated in children with high CU traits, one might speculate that a similar neuronal mechanism centered on the amygdala might be underlying this phenomenon [17]. 

In contrast to Experiment 1, where the initial fixation on facial expressions was determined passively by unpredictably shifting stimuli on the screen, attention to the eye region was increased purposefully (rather than passively) in the second Experiment. We conducted this pilot intervention in the US using a juvenile offender sample which, in terms of emotional and behavioral problems, was similar to that of the UK alternative school that was examined in Experiment 1. 

## 2. Experiment 1

In the first study, we tested whether the fear recognition deficits observed in those high on CU traits may be explained by problems in reflexively attending to the salient facial features that are diagnostic of fear in faces (i.e., the eyes). We recruited youths (mainly boys) from multiple English alternative schools that serve those who have been expelled from other institutions for emotional and behavioral problems. In this experiment, our first aim was to determine whether children high on CU traits show a selective deficit in reflexive gaze shifts toward the most diagnostic region of fearful faces, specifically, the eye region. Youths with CU traits and impulsivity/conduct problems (ICP) were expected to show deficits that are not shown by those with elevated levels of CU traits only. We examined school case records to code for and examine ICP and their role in gaze shift behavior toward the eye region when participants looked at fearful faces, with CU traits as a moderator. We also aimed to replicate the fear recognition deficit related to CU traits.

### Directing Eye Gaze

A secondary aim of Experiment 1 was to determine whether fixating on the eye region of faces by manipulating the initial fixation can reduce the fear-recognition deficit in children with CU traits. Based on the relevant literature, e.g., [2], we hypothesized that those with high CU traits would show a fear-recognition deficit in the condition when the mouth region of the face was presented at the fixation point; however, we further predicted that this deficit would be reduced in the condition when the eye region was presented at the fixation point. However, not all studies find that this type of passive attention to fearful eyes is helpful for those with CU traits. In one study, cueing fearful eyes with a blue dot that flashed at various points on the face failed to activate the amygdala for antisocial youths high on callous traits when it flashed in the region of the eyes [21]. However, findings may depend on comorbid behaviors. Sebastian and colleagues [21] found that the eye-gaze deficits were evident in youths who were high on both antisocial traits and callous traits. Therefore, in the present study, we examined the interaction of ICP and callousness, separately, as well as the interaction with the other facets of CU traits (i.e., uncaring and unemotional).

## 3. Materials and Methods

### 3.1. Participants

Participants (*n* = 73; 61 males) were youths from four special schools (Emotional and Behavioural Difficulty Schools) in England and were between 11 and 16 years of age (*M* = 14.0 years, *SD* = 1.4 years). Four schools in the North East of England were included in recruitment process. Each school varied on recruitment success rate (71%, 87%, 60%, 42%). Participants were predominantly White British (96%). Based on school records, 23% had lived in alternative care, 52% had a diagnosis of Attention Deficit Hyperactivity Disorder (ADHD), three with co-morbid Oppositional Defiant Disorder, two with co-morbid Conduct Disorder, 5% with Autism Spectrum Disorder (ASD), and 5% with depression. Staff were asked about any contact with the police; 41% of children had had contact with the police for crimes such as stealing a car.

Parents or legal guardians were first contacted by a school official (usually the Head or the Assistant Head) to gain permission for the researchers to talk to them on the phone. The researchers were handed the phone if the guardians agreed to hear about the research. This way, researchers were not given the guardian’s phone number. Researchers then told the guardian about the study and asked to record the verbal consent, stating that the written consent form would be posted following the phone call or left at the school for guardians to collect. Once consent was received from guardians, we asked children individually if they wanted to participate, by bringing them to a private room in the school and providing full information about the study and voluntariness of participation.

### 3.2. Measures

#### 3.2.1. Callous-Unemotional Traits

The Inventory of Callous-Unemotional traits [22] is a self-report which includes 24 items, such as “I am concerned about the feelings of others” that are rated on a four-point Likert scale from 0 (Not at all true) to 3 (Definitely true), which has demonstrated good reliability [23]. In the present study, the ICU total score yielded good internal consistency (Cronbach’s α = 0.82). Of the three validated facets, the callous (α = 0.75) and uncaring facets (α = 0.83) showed good, but the unemotional facet poor reliability (α = 0.52). Mean and dispersion (*M* = 27.6; *SD* = 9.3) of the ICU were similar to other studies. For example, they were similar to youths (70% male) from a residential facility program in the United States (*M* = 25.74, *SD* = 7.95; Lui et al. [24]) and to individuals from a community school sample in the United Kingdom who were in the same age range of 11–16 years (male: *M* = 25.25, *SD* = 7.90; female: *M* = 21.76, *SD* = 9.4; Muñoz et al. [25]).

#### 3.2.2. Anxiety

The Behavior Assessment Scale for Children, 2nd edition (BASC-2; [26]) is a standardized and norm-referenced rating scale that is widely used among clinicians and researchers. The BASC-2 measures emotional and behavioral functioning and self-perceptions of children and adolescents. There is abundant support for the reliability and validity of the BASC-2 in adolescent samples [27]. We used the self-report subscale of anxiety and created T-scores based on norm-references [26]. The range in the present study was 34 to 78 with an average of 53.54 (*SD* = 11.64).

#### 3.2.3. Face Perception Task

The experimental task used was comparable to a previous study by Boll and colleagues [8] and allows for measuring automatic and more sustained aspects of visual orienting during face processing. It was based on a 2 × 2 × 4 design with the within-subject factors presentation time, initial fixation and emotional expression. Male and female faces with either a fearful, angry, happy, or neutral expression were selected based on validation studies from several established data sets (KDEF: http://www.emotionlab.se/resources/kdef, accessed on 12 March 2011; NimStim Face Stimulus Set: http://www.macbrain.org/resources.htm, accessed on 12 March 2011; Pictures of facial affect: Ekman and Friesen, [28], and the FACES database: Ebner et al. [29]). All images were converted to grayscale, cropped with an elliptic mask to hide hair and ears, and the cumulative brightness was normalized across images. Stimulus selection was accomplished individually for each participant and involved drawing a random sample of 40 individual faces (20 male, 20 female) for each emotional expression from the whole stimulus pool that consisted of 502 faces in total. The selected faces were assigned to the experimental conditions in a balanced way (i.e., the male/female ratio was constant across conditions) and the resulting 160 trials were split into three experimental sessions with 55, 55, and 50 trials, respectively, with randomized trial order. Each trial started with a fixation cross in the center of the screen that was shown for 1 s. Subsequently, a face was presented on the computer screen for either a very brief time period (150 ms) or for a longer duration (3000 ms). For brief stimulation durations, a uniformly gray screen followed the face presentation for 1850 ms. In order to control for the initial fixation, faces were unpredictably shifted either up or down on each trial such that participants either initially fixated on the mouth or the eye region. Eye gaze was monitored in each condition using an EyeLink 1000 (SR Research Ltd., Ottawa, ON, Canada), which was positioned 56 cm in front of the participant’s eyes. Head position was fixed using a chin rest. Once the face had been presented participants were asked to verbally report which facial expression they had been presented with. After the response, the fixation cross reappeared on the screen for an intertrial interval with a randomly chosen duration of 1 to 3 s.

The short duration condition was included in the design to ensure that participants could identify the given facial expression without being able to change the point of their fixation during stimulus presentation. Any automatic saccade in response to the stimulus presentation would occur when the picture already disappeared from the screen. By contrast, within the larger presentation time, participants were able to make several fixation changes during stimulus presentation to visually scan the face in detail. In the current article, we only focus on the short presentation time (150 ms), because this condition is of greatest interest to examine reflexive attentional orienting which is probably mediated by the amygdala [14].

From the recorded eye-tracking data, we quantified the proportion of saccades towards the facial feature that was presented in the visual periphery (i.e., not at fixation). As in our previous studies [8,18], we first identified saccades in the recordings using velocity and acceleration thresholds of 30°/s and 8000°/s^2^, respectively. Trials with saccades >1° or eye blinks during a baseline period of −300 to 150 ms relative to stimulus onset were declared as invalid and removed from further analyses. Subsequently, we analyzed the direction of the first saccade with a vertical amplitude >1° that occurred within a time interval of 150–1000 ms after stimulus onset. These saccades were classified according to their direction and they were coded as downward or upward fixation changes depending on whether the eyes or the mouth were initially fixated. Trials with vertical amplitudes of less than 1° were coded as remaining on the initially fixated area. The total numbers of saccades across trials were divided by the number of valid trials per experimental condition to determine proportions.

### 3.3. Procedure

The experimental task took place in a quiet room within the school. Self-report ICU was completed by the participants prior to the experiment to accommodate a stabilization period for physiological measures (which were part of a different study). All questionnaires were read to participants to eliminate effects of reading ability. Following the questionnaires, participants were shown the eye-tracker, and the headrest was clipped to a table ensuring it was comfortable for participants. After practice trials, participants completed the eye gaze task. Participants were told to verbally respond with the emotion they thought the face was displaying, and the researcher made a corresponding keyboard button press to log the responses. The researcher sat where they could not see the faces on the screen to put the participants at ease that their performance would not be evaluated. Researchers logging the answers were unaware of the CU or ICP status of participants. All participants received a chocolate bar for completing the study, and this was cleared with the Head of school to ensure it was balanced with other dietary requirements. School records were examined following all experimental protocol to code impulsivity and conduct problems (i.e., ADHD and/or conduct disorder or oppositional defiant disorder). Those with autism spectrum were categorized as not having ICP. Ethical approval was granted by Durham University (#12.31) and the local school board.

## 4. Results

### 4.1. Are CU Traits and ICP Related to Lower Proportion of Reflexive Saccades toward the Eye Region When the Initial Fixation Is on the Mouth?

Analyses were conducted in JASP 0.14.1 [30]. Missing values were due to computer malfunction (*n* = 4), participant drop-out partway through testing (*n* = 3), or refusal to complete the questionnaires (*n* = 7). Complete eye-gaze data was available for 61 participants and emotion recognition accuracy for 64 participants. ICU scores were available for 60 participants. 

Figure 1 shows the full sample’s gaze shift, which is similar to previous studies with adults [9]. Thus, as expected, children moved their gaze to the eye region more often when initially fixated on the mouth for anger, fear, and neutral faces, but the opposite was true for happy faces where they shifted to the mouth more often.

We first tested whether CU traits were related to the number of eye-tracking valid trials, but these were not significantly related, Kendall’s tau B = −0.016, *p* = 0.869. No gender differences were found on CU traits and anxiety, t(57) = −0.55, *p* = 0.582, Cohen’s d = −0.19 and t(57) = −1.72, *p* = 0.092, Cohen’s d = −0.60, respectively. 

To test the hypothesis that children high in callous-unemotional traits show a selective deficit in reflexive gaze shift toward the most diagnostic region (i.e., eyes) of fearful faces, non-parametric correlations were examined to handle proportional data. Table 1 shows that CU was not significantly related to the proportion of saccades to the eye region of fearful faces. In other words, children high on CU traits did not show a significant deficit in reflexive gaze to the eye region of fearful faces. Similar non-significant effects were observed for the other emotions. This would suggest that children with high CU traits do not show deficits in shifting their attention towards salient features of the face in determining emotions; yet, almost all correlations are negative, possibly suggesting children with CU traits fail to reflexively shift their attention from any point that was initially fixated, even though the choice of fixation was varied systematically. Because CU traits are associated with anxiety, we included a measure of anxiety, which was also nonsignificantly associated with gaze shifts.

We next examined the impulsivity/conduct problem groups which were composed of 30 youths with ADHD. Those with comorbid ASD were included in the group without ICP, comprising 43 of the youths. Of note, the ICP groups did not differ on CU traits, t(58) = 1.09, *p* = 0.279, Cohen’s d = 0.28, or anxiety, t(57) = −0.50, *p* = 0.620, Cohen’s d = −0.13. To test whether children diagnosed with ICP had problems in reflexively shifting attention to the eyes of fearful faces, we conducted t-tests to examine if those with ICP made fewer saccades to the eye region of faces when the mouth was at fixation. We found significant effects for the proportion of saccades toward the eyes for fearful faces, t(61) = 2.47, *p* = 0.016, Cohen’s d = 0.63. Figure 2 shows that ICP are associated with a reduced reflexive orienting towards the salient features (i.e., the eyes) that assist with diagnosing fear in faces. 

The comparable t-test for saccades toward the mouth when the eyes were fixated in fearful faces was not significant, t(61) = −0.72, *p* = 0.475, Cohen’s d = −0.184. All other t-tests examining happy, neutral, and anger were non-significant for reflexive shifts to the eyes when the mouth was fixated t(62) = 1.72, *p* = 0.090, Cohen’s d = 0.44, t(62) = 0.945, *p* = 0.348, Cohen’s d = 0.241, t(62) = 1.28, *p* = 0.205, Cohen’s d = 0.33, respectively.

### 4.2. Are CU Traits Related to Fear Recognition Deficits?

First, we examined whether reflexive saccades toward the eye region were related to better recognition. None of the correlations were significant for fear (0.05), anger (0.09), neutral (0.11), and happy (−0.09). The correlations between CU traits and accuracy in emotion recognition are noted in Table 2. As expected, children high on CU traits performed more poorly in labelling fearful faces accurately; however, the fear recognition deficit was only significantly evident for those higher on CU traits when the mouth of the fearful faces was fixated. A Williams test for testing differences between dependent correlations was run to examine the hypothesis that children who score highly for callous-unemotional traits show a fear-recognition deficit specifically when the mouth region and not when the eye region of the face is primed. The results were non-significant, thus there was no significant difference regarding the association between ICU scores and fear recognition across the mouth-fixation condition and the eye-fixation condition, t(50) = 0.53, *p* = 0.596. In examining whether ICP groups differed on emotion recognition accuracy, none of the *t*-tests were significant (ts ranging from −0.95 to 1.72, ps ranging from 0.205 to 0.766).

### 4.3. Do CU Traits Moderate the Relation between Impulsivity/Conduct Problems and Reflexive Gaze toward the Eye Region of Fearful Faces?

Given that youths with ICP showed less shifts to the eye region when viewing fearful faces, in an exploratory analysis, we examined whether CU traits moderate this link. Sebastian et al. [21] found that cueing fearful eyes with a blue dot that flashed around the face failed to activate the amygdala for antisocial youths who were also high on the callous facet of CU traits. Specifically, in line with their findings, we looked at the combination of impulsivity/conduct problems and CU traits. Thus, we tested the interaction between mean-centered callousness, uncaring, and the unemotional facets of CU traits and ICP (CU facets x ICP). In a hierarchical regression, we included the continuous variables of callous, uncaring, and unemotional traits and the categorical variable of impulsivity/conduct problems [1 = ICP, 0 = No ICP]) in the first step as predictors of the proportion of gaze shifts when the mouth of fearful faces was first fixated. The three interaction terms were included on the second step, to test for a potential moderation.

The first step of the regression was not significant in explaining variance in gaze shifts when the mouth was fixated onto fearful faces, F(4, 48) = 1.36, *p* = 0.261, R^2^ = 0.10. The second step of the model, when all the predictors and the multiplicative (interaction) terms were included, was also nonsignificant, F(7, 45) = 2.22, *p* = 0.050, R^2^ = 0.26, and the ΔR^2^ of 0.16 was significant, ΔF(3, 45) = 3.13, *p* = 0.035. Additionally, as expected based on Sebastian et al. [21], the beta for the interaction between callous traits and ICP was significant while controlling for the other facets of the ICU and their interactions with ICP, beta = −0.07, SE = 0.025, β = −0.56, t = −2.76, *p* = 0.008. VIF statistics indicated no issues of multicollinearity (range 1.09 to 3.26). To examine the form of this interaction, PROCESS version 2.13 [31] for IBM SPSS v.24 was used. The moderation model with ICP as the statistical predictor and eye gaze shifts when fearful mouths were at fixation was used while centering the data and controlling for the other facets of the ICU. Figure 3 was produced with the Plot function in PROCESS to show the form of the interaction, plotting the values of eye gaze shift in relation to ICP status for those at the mean (average), and ±1 *SD* of callousness. We looked at the Johnson–Neyman significance regions of the moderator, which calculates the range of values for the observed data for which the interaction is significant [31]. The results indicated that for those at the mean (approximately) and higher, the slope is significant. That is, the interaction becomes significant just below the mean (at −0.0635) and is significant at all values above the mean and no values below −0.0635. The Johnson–Neyman points indicate that the association stays significant up to the highest value in our sample, which is about 3 *SD* above the mean (13.11). Nevertheless, it is worth noting this was a small effect size, and that the other facets had variable betas, likely adding noise and leading to a non-significant change in R^2^, beta (uncaring) = 0.04, SE = 0.03, β = 0.40, t = 1.78, *p* = 0.081 and beta (unemotional) = −0.04, SE = −0.03, β = −0.21, t = −1.19, *p* = 0.239. Therefore, we encourage further replication. 

In sum, the current analyses indicate that impulsivity/conduct problems relate to deficits in gaze shifting when a fearful mouth is fixated first, but total CU traits do not. Callousness appears to be related to the deficits in gaze shift associated with ICP. CU traits are related to deficits in reading fearful facial expressions when the mouth is fixated first.

## 5. Discussion

Taken together, the results from Experiment 1 show those with high levels of CU traits fail to take in and process fear in other people’s faces accurately, although the association with fear recognition deficits might depend on the initial fixation (i.e., the eyes). The emotional deficits shown in this study seemed to be based on different factors related to empathizing with others: the problems appear to be attentional for youths with impulsivity/conduct problems, given that they failed to attend to fearful eyes. In contrast, the deficits in recognizing fear were related to CU traits. Further, there was evidence of emotion processing deficits for youths with a combination of callous traits and impulsivity/conduct problems. Youths high on ICP who are also high on callous traits shifted their gaze less toward fearful eyes when initially focused on the mouth. Only the callousness facet was significantly associated with decreased gaze shift. As these analyses were exploratory, it is important that further research be conducted before we can confidently state that the combination of CU and ICP result in problems in reflexive orienting or shifting to the eyes in order to understand fear. 

The findings from Experiment 1 indicate that shifting the initial fixation to the mouth region of fearful faces yields problems in fear recognition in youths with high CU traits. However, based on findings by Dadds and colleagues [2], when instructed to “look at the eyes” there is evidence of more accurate fear recognition. It is not known, though, whether an intervention aimed at increasing purposeful (rather than passive) attention to the eye region will lead to changes that continue beyond an initial trial. Experiment 2 examines whether such an intervention has effects over a longer time interval.

## 6. Experiment 2

Despite growing support of amygdala-centered deficits in psychopathy, e.g., [17], this body of evidence is still incomplete among juveniles. Using an adjudicated sample of adolescents that included youth with elevated CU traits, this pilot test aimed to investigate whether these juveniles had sustained improvements in their recognition of fearful faces after being administered an intensive learning intervention that included repeated eye-gaze instruction to specifically “look at the eyes” of face stimuli. This sample was, therefore, suited to examine whether adolescence may be a developmentally sensitive period in which there is some malleability of fear recognition deficits. Stability of psychopathic traits notwithstanding, examining potential malleability in youth samples may point to periods in which intervention could be effective for amelioration of emotion recognition deficits contributing to fear blindness. 

A secondary aim of the study was to investigate whether this hypothesized learning process generalized to improvements in empathic responding and attention to other emotional cues of interest. Although empathy and fear recognition are theorized to be linked, such that fear recognition is a part of emotion understanding, the role of learning has not been specifically demonstrated in the context of this link. Deficits in attentional processing may impair empathic responding because such deficits would diminish the salience of emotional cues [32]. In Experiment 1, we showed that youths with impulsivity/conduct problems (ICP) and callous traits made fewer reflexive shifts to change their attention (measured as eye-gaze) to the eyes of fearful faces, which would have aided their understanding of emotions. We argue that learning to recognize fear in others may be helped by encouraging attention to emotional cues such as those associated with fear.

Consistent with previous works [2,7,33,34], we hypothesized that CU traits and ICP would be associated with fear recognition deficits, low levels of empathy, and low levels of attention to threat cues. However, we expected that CU traits would predict significant changes in fear recognition, as indexed by increased accuracy across waves in an emotion recognition task, over and above ICP, and within the group receiving repeated eye-gaze instruction. Specifically, individuals with elevated CU traits in the repeated eye-gaze instruction condition were predicted to show greater improvements in fear recognition as compared with those who received the eye-gaze instruction only once. We expected to see the largest effect among these youth, because youth with higher levels of CU traits were expected to have poorer fear recognition at baseline as a consequence of their failure to attend to the most salient cues—the eyes [2,35]. By the same rationale, we predicted that higher levels of CU traits would predict significant change in attention to distress and threat cues, as indexed by significantly reduced response times to non-facial distress and threat stimuli, relative to baseline. Finally, we expected that CU traits would also predict increased empathic responding as indexed by significant changes in measures of empathy, when receiving the instruction more than just once.

## 7. Materials and Methods

### 7.1. Participants

Participants *(n* = 34; 29 male) were youths from a juvenile detention center in the Northeastern United States. Participants comprised 2 evenly distributed experimental groups differing on exposure to the learning paradigm. The self-reported racial distribution of the sample mirrored the state (i.e., 5% minority) from which they were drawn (90% White, 10% African-American, bi- or multiracial). 

Inclusion criteria specified youths: (1) ages 14 to 17 at baseline, (2) available for three consecutive waves of data, (3) functioning within the normal range of intelligence as indexed by academic performance. This determination was made in consultation with academic classroom teachers and staff in the detention center school program. Exclusion criteria included evidence of (1) intellectual impairment that might limit ability to validly complete the protocol, and (2) limited cognitive ability, including any of the following: intellectual disability, pervasive developmental disorders, selective mutism, organic mental disorders, schizophrenia, other psychotic disorders, or inability to give informed, written assent to participate in research. A total of 6 youths were excluded. Three were excluded because they were unexpectedly transferred to another placement prior to Wave 2, one youth withdrew during the second wave because they were no longer interested in participating, and 2 youths declined to give assent after caseworker (legal guardian) consent was obtained. 

### 7.2. Measures

#### 7.2.1. Anxiety and Depression

The Revised Child Anxiety and Depression Scale (RCADS) [36] is a 47-item questionnaire designed to assess symptoms corresponding to selected DSM-IV anxiety disorders and major depression. Using a 4-point scale ranging from “never” to “always,” respondents rated statements about themselves (e.g., “I feel sad or empty,” “I have problems with my appetite”). Factor analyses have supported 6 subscales of the RCADS, differentiating DSM-IV disorders. These include Separation Anxiety Disorder (SAD), Social Phobia (SP), Generalized Anxiety Disorder (GAD), Panic Disorder (PD), Obsessive-Compulsive disorder (OCD), and Major Depressive Disorder (MDD). Scores on the RCADS can be summed to yield a Total Anxiety Scale (sum of the 5 anxiety subscales) and a Total Internalizing Scale (sum of all 6 subscales). The RCADS has been found to have convergent and discriminant validity with other depression and anxiety measures and has generally displayed greater internal consistency and test-retest reliability (Cronbach’s α = 0.71–0.85) than other similar measures [36]. For the current study, the internal consistency was excellent (Total Internalizing Scale: Cronbach’s α = 0.91, Total Anxiety Scale: Cronbach’s α = 0.89).

#### 7.2.2. Impulsivity/Conduct Problems (ICP)

The Antisocial Process Screening Device (APSD) [37] was designed to identify a subgroup of antisocial youth displaying traits similar to those of adult psychopaths in adjudicated, clinical, and community samples [38]. The Impulsivity conduct problems scale (ICP) of the APSD is a 5-item scale that measures antisocial behavior (e.g., “Does risky things”), impulsivity and recklessness (e.g., “Acts without thinking”), and a tendency to be easily bored (e.g., “Gets bored easily”, [39]). These traits are placed in a rating scale format with ratings of 0 (“not at all true”), 1 (“sometimes true”), or 2 (“definitely true”). The reliability for this scale has been estimated at 0.77 [40], and it shows independent predictive utility for antisocial outcomes [41,42]. For the current study, the ICP demonstrated an acceptable level of internal consistency (Cronbach’s α = 0.72).

Because the ICP scale consists of only 5 items, the Eysenck Impulsiveness Scale (EIS) [43] was also used to assess impulsivity. It is one of three scales (Impulsiveness, Venturesomeness, and Empathy) that is part of a modified version of the Junior Impulsiveness Questionnaire. The EIS consists of 23 yes-no questions assessing cognitive and behavioral impulsivity. Scores range from 0 to 23. Some examples of questions include “Would you enjoy gambling?” and “Do you usually think carefully before doing anything?” The reliability of the EIS is estimated at 0.84 [43], and the EIS is valid for assessing impulsivity among youth with severe conduct problems [44]. However, the EIS evidenced poor internal consistency in the current study (Cronbach’s α = 0.51).

#### 7.2.3. Callous-Unemotional Traits

Similar to Experiment 1, CU traits were measured with the Inventory of Callous-Unemotional Traits (ICU, [38]. The ICU demonstrated good internal consistency in this study (Cronbach’s α = 0.88), and there was an even distribution of ICU scores among participants.

#### 7.2.4. Empathy

The Interpersonal Reactivity Index (IRI, [45,46] is a 28-item measure of empathy rated on a 5-point scale ranging from “does not describe me very well” to “describes me very well.” Factor analyses have supported 4 subscales of empathy as measured by the IRI (i.e., Fantasy Empathy (FE), Perspective Taking (PT), Empathic Concern (EC), and Personal Distress (PD)); only the 3 subscales that have been widely used in similar samples were used in the current study. That is, the Fantasy Empathy scale has been found to be unrelated to the processes and outcomes of interest; thus, those data have not been included in any of the current study’s analyses. The PT subscale contains items related to the tendency to spontaneously attempt to view situations from others’ points of view (e.g., “I believe that there are two sides to every question and try to look at them both”). The EC subscale taps the degree to which an individual feels compassion, warmth, and concern for another person’s misfortune (e.g., “I often have tender, concerned feelings for people less fortunate than me”). The PD subscale measures the tendency to personally feel apprehensive, fearful, and uncomfortable as a result of viewing the negative experiences of others (e.g., “When I see someone get hurt, I tend to remain calm”). The IRI has displayed acceptable internal consistency and evidence of predictive and convergent validity (Cronbach’s α = 0.70–0.78, [46,47]. Across the waves of data collection, the PT subscale exhibited internal consistency ranging from questionable to good (Wave 1: Cronbach’s α = 0.64, Wave 2: Cronbach’s α = 0.78, Wave 3: Cronbach’s α = 0.87). The internal consistency demonstrated by the EC subscale ranged from acceptable to good (Wave 1: Cronbach’s α = 0.83, Wave 2: Cronbach’s α = 0.76, Wave 3: Cronbach’s α = 0.83). The PD subscale had internal consistency ranging from questionable to good (Wave 1: Cronbach’s α = 0.77, Wave 2: Cronbach’s α = 0.69, Wave 3: Cronbach’s α = 0.84).

#### 7.2.5. Attention to Emotional Cues

The Emotional Pictures Dot-Probe Task (EDPT) [48] is a pictorial variation of the word task developed by MacLeod, Mathews and Tata [49]. The task is used to index attentional orienting to specific categories of emotional stimuli and has been widely used in evaluating the relation between anxiety and attentional orienting in children [50] and adults [51]. Further, in children it has been used in investigations of the relation between emotional processing and aggressive behaviors [52] as well as in an investigation of emotional processing and psychopathic traits [53]. Typically, this task pairs neutral and emotionally salient pictures from the International Affective Picture System [54]. For the purposes of the current study, neutral (e.g., a table), positive (e.g., a smiling, happy child), threat (e.g., a snarling dog), and distress (e.g., a crying child) pictures were used. In order to address the insufficient number of threat pictures in the IAPS set, additional pictures were added that mirrored the IAPS content. A small pilot study was conducted (*n* = 5) to ensure that these added pictures adequately fit the criteria of being threatening. In terms of valence and arousal, the additional pictures were internally consistent (Cronbach’s α = 0.92 and Cronbach’s α = 0.85, respectively). In addition, the internal consistency of all of the threat pictures used was excellent (Cronbach’s α = 0.94). However, the internal consistency of the distress pictures was questionable (Cronbach’s α = 0.64). 

Following previous applications of this task [55], the task consisted of 1 practice block (16 neutral-neutral picture-pairs) followed by 3 experimental blocks (24 picture-pairs, each). Each picture-pair trial consisted of a fixation cross that appeared in the center of a computer screen for 500 ms followed by the simultaneous presentation of 2 picture stimuli centered and located immediately above and below the location of the fixation cross for 500 ms. An asterisk (i.e., dot-probe) immediately followed the test stimuli, appearing in either the top or bottom picture location for 500 ms. Participants were then required to select a key on the keyboard corresponding with the location on the screen (up or down) where the dot-probe appeared, as quickly and as accurately as possible. The location of the dot-probes and the pictures were counterbalanced across test trials in order to ensure equal placement of the stimuli in the top and bottom locations. 

Failure to press a key within 5000 ms resulted in the response being recorded as incorrect. Because this lack of responding indicated that the participant was not paying attention to a specific stimulus pair, incorrect responses were not included in subsequent calculations. In addition, response times less than 100 ms were not included in calculations because they were considered outliers resulting from program error. Response times falling more than 3 standard deviations above or below the mean were also considered outliers. Notably, in the current study, no data had to be excluded based on these stipulations. Participants were required to reach at least 70% accuracy on the practice block in order to be considered trained on the task and adequately prepared to participate in the experimental blocks; all of the participants met this requirement. However, across the experimental blocks of the three waves of data collection, four participants got less than 70% of their responses correct on any one of the categories of picture-pairs (e.g., distress-neutral pictures).

The outcome variable was the mean facilitation index score for distress pictures. Participant response time, as indexed by the time between when the dot-probe appeared and when the corresponding key-press occurred, was recorded in milliseconds and used for the calculation of the facilitation index scores. Specifically, the average response time to dot-probes located in the same location as distress pictures in distress-neutral picture pairings was subtracted from the average response time to dot-probes located in the same location as neutral pictures in neutral–neutral picture pairings. Potential location effects were controlled for by calculating facilitation scores that compared latency to emotional and neutral dot-probes in the same location using the following formula: facilitation = ½ × [(neutral only | probe up—emotional picture up | probe up) + (neutral only | probe down—emotional picture down | probe down)], based on MacLeod and Mathews [56]. Thus, positive facilitation scores indicated normal response to emotional pictures, because emotional pictures tend to garner more attentional focus and elicit quicker responses when the dot-probe replaces them. Data were screened to ensure that no facilitation scores fell more than three standard deviations above or below the mean. 

#### 7.2.6. Facial Emotion Recognition and Learning

The University of New South Wales (UNSW) Facial Emotion Task (FACES; [57]) is a measure of emotion recognition accuracy. It consists of a PowerPoint presentation divided into three blocks displaying happy, sad, angry, fearful, disgusted, and neutral facial expressions. In the first block, the free-gaze condition, seven faces (three adult, two adolescent, and two child) displaying one of the six emotions were displayed randomly. The faces were presented one at a time for 2 s. Participants were asked to select the emotion portrayed from a list of the six emotions on a record form, and the researcher recorded the responses. Since the first block was used as a baseline measure of emotion recognition accuracy, participants did not receive feedback on their responses. In the second block, the eye-gaze instruction block, participants were instructed to focus on the eyes of the facial stimuli presented. To increase compliance with the eye-gaze instruction, a slide with an X positioned in the place of the eyes appeared for 1 s before each facial stimulus appeared. Participants selected the emotion shown on the record form. Only the three adult faces were shown in the eye-gaze instruction condition. During this condition, participants received feedback on their responses. As in the first block, in the third block (i.e., the emotion recognition condition), participants indicated which emotion was depicted in each of the facial stimuli presented. They received neither explicit instruction to look at the eyes of the facial stimuli presented nor additional feedback on their responses. An accuracy score was calculated for each emotion, for each block. Across adolescent and adult samples, FACES has been used to distinguish between different response patterns for individuals with elevated and low levels of psychopathic traits [2]. Moreover, previous work has indicated that the instructional intervention has increased fixation on the eyes and fear recognition for that trial [2].

As noted in the Procedure section below, data were collected in three waves (see Figure 4 for a summary of the waves of data collection). Specifically, the free-gaze condition was ordered as first in all blocks for all groups throughout the three waves in order to avoid contamination effects of the eye-gaze instruction (see Figure 5). However, groups varied on whether they received the eye-gaze instruction condition during the second block of a Wave. Specifically, one group (intervention group) received the eye-gaze instruction during the second block of both Wave 1 and Wave 2. The other group (control group) received the instruction during the second block of Wave 1, only. Both groups received the free-gaze condition in the third block of all three waves, in order to investigate the short-term effects of the learning task. In Wave 3, all participants received the free-gaze condition, in order to investigate the sustained effects of the learning task.

### 7.3. Procedure

Detention center staff members contacted research team members when it was determined that a youth was eligible to participate. Since the state has legal custody of youths in detention centers, consent for eligible youths was obtained from Department for Children and Families caseworkers prior to notifying youths of their eligibility to participate. Caseworkers and participants were advised of the project’s nature, goals, potential benefits and risks, and compensation for participation. Each participant had a private meeting with project staff both to obtain consent and to complete the protocol during each wave of data collection. Because incarcerated youths are a vulnerable population, a representative from the State Juvenile Defender’s Office was available on site or by phone at all times to ensure that eligible youths had all questions answered by a neutral party and could freely agree to participate or decline participation.

With the exception of the ICU, which is typically administered to all center residents at intake, data (i.e., demographics, questionnaires, tasks) were collected in face-to-face interviews. A laptop computer was used to record the data collected. Participants were asked to report their current age, date of birth, race and ethnicity, grade in school, age of onset of antisocial behavior, and age of first arrest, orally. 

The protocol was administered during three waves of data collection, occurring between five and ten days apart (*M* = 6.38 days, *SD* = 1.68 days). The initial (i.e., Wave 1) session took approximately 60 min to complete. In this session, participants completed the (1) IMP, (2) EIS, (3) RCADS, (4) IRI, (5) EDPT, and (6) FACES. Subsequent sessions (i.e., Wave 2 and Wave 3) took approximately 40 min and consisted of the (1) IRI, (2) EDPT, and (3) FACES. The order of the assessments was counterbalanced during each wave. Following past work with this population [42,58,59], assessments were administered in a face-to-face interview using standardized electronic forms on a laptop computer to eliminate bias related to differences in reading ability or in the understanding of study items. Additionally, to aid with comprehension and to increase the accuracy of responses, the interviewer provided participants with color-coded placards for each measure indicating response scales and word anchors for possible responses. Each placard listed the response choice numbers (e.g., 0, 1, 2, 3) with word anchors for the rating scale (e.g., not at all true, somewhat true, very true, definitely true). The interviewer entered participant responses on the computerized forms, and these responses were automatically added to a database for each measure. Accuracy scores for the FACES task were recorded by research staff on a record form and calculated and entered into the database after the interview was completed. 

Participants were compensated with gift certificates to local vendors and online merchants after completing each data collection wave. They received a $10 gift certificate at the completion of both the first and the second waves of data collection. Participants were given two additional $10 gift certificates upon completion of the third wave of data collection. Thus, participants who completed the full protocol received four gift certificates, worth a total of $40. The protocol was approved by both the University and the State Human Services IRBs.

## 8. Results

### 8.1. Preliminary Analyses and Data Analytic Strategy 

#### Missing Data

Participant retention across the waves of data collection was adequate, with 30 of the 34 participants (88%) completing at least two of the three waves. Only these 30 participants (27 male) were included in analyses of study hypotheses. Twenty-eight participants (82% of the 34 participants) completed all three waves of data collection. The remaining participants had some missing data across waves. 

Data were missing at random (MAR); consequently, they were handled with multiple imputation (MI) [60,61,62] for descriptive statistics and with Full Information Maximum Likelihood Estimation (FIML) for primary analyses. MI provides unbiased, generalizable estimates of missing values and standard errors [63] and is considered a best practice statistical approach for handling missing data [62,64]. Both descriptive statistics and correlations were conducted following MI. However, linear mixed modeling (LMM) using FIML was used to address missing data when conducting primary analyses aimed at examining changes in dependent variables over the three waves of data collection. The use of LMM analyses was supported because (1) they adequately account for correlated data across multiple time points, (2) they allow for tests of mean differences between groups, (3) they provide valid parameter estimates and significance tests when there are missing data in repeated measures analyses [65,66,67], and (4) estimates of models were similar using both MI and LMM. Thus, reported descriptive statistics and zero-order correlations reflect use of multiple imputation for missing values, and LMM results reflect use of maximum likelihood estimation for missing values.

LMM analyses addressed this study’s aims by determining whether fear recognition, response to both fearful stimuli and threat stimuli (as measured by the EDPT), and empathy trajectories differed for the two study groups across the waves of data collection. Most importantly, these analyses assessed whether the trajectories could be accounted for by differences in the level of CU traits. For the current study, time was coded as 0 (Wave 1), 1, (Wave 2), and 2 (Wave 3). Therefore, the intercepts reflect levels of the dependent variables at Wave 1, and linear slopes reflect change in the dependent variables from Wave 1 to Wave 3. Because interactions were predicted and tested, predictors were mean centered.

Although other models were considered (i.e., random intercept, random slope, and random intercept and slope), preliminary analyses comparing models established that fixed effects models provided the best fit to the data in the hypothesized models. Specifically, the random intercept model and the random slope model resulted in essentially no change in fit, and adding both components simultaneously resulted in a significantly worse fit as compared with fixed effects models. Additionally, the unstructured covariance structure was deemed most appropriate. Thus, unstructured fixed effects models were used for all analyses. 

### 8.2. Descriptive Statistics

To address the first hypothesis, that CU traits and ICP would be associated with fear recognition deficits, low levels of empathy, and low levels of attention to fear and distress cues, bivariate correlations were examined. Correlations, means, and standard deviations for main study variables are presented in Table 3. Correlations, means, and standard deviations for all study variables by group are available by request from TRS.

Correlations were generally similar for both groups, though there were a few differences in these associations. 

For example, there were differences in correlations between Impulsivity (as measured by the APSD ICP scale) and study variables. Specifically, for the intervention group, ICP was negatively associated with fear recognition at the first wave (−0.53, *p* < 0.05), only; whereas, for the control group, ICP was positively associated with fear recognition at all three waves of data collection (0.67, *p* < 0.01; 0.69, *p* < 0.01; 0.69, *p* < 0.01, respectively). In addition, for the control group, ICP was positively correlated with facilitation to threat in the second wave of data collection (0.55, *p* < 0.05). There were no significant correlations between facilitation scores and ICP or CU traits in the intervention group. Finally, for the intervention group, ICP was negatively associated with perspective taking at all three waves (−0.58, *p* < 0.05; −0.61, *p* < 0.05; −0.55, *p* < 0.05, respectively); however, for the control group, ICP was negatively associated with perspective taking at the second (−0.58, *p* < 0.05) and third waves (−0.69, *p* < 0.01), only. Again, the pattern of association was similar across all waves, with slightly stronger correlation in the control group. ICP as measured by the EIS was not significantly associated with any study variables of interest. ICP was used as a covariate in the primary analyses. 

Study groups appeared to be essentially equivalent at baseline, suggesting that randomization was effective for the measured variables and that differences observed in later data collection waves were most likely due to the intervention implemented. No significant differences between the intervention group and the control group were detected on Wave 1 outcome variables (fear recognition accuracy, response time to distress stimuli, response time to threat stimuli, and empathy measures). Additionally, there were no significant differences between predictor variables (CU traits and ICP). Gender was included in LMM analyses as a covariate to minimize any biasing effects the small number of female participants may have introduced. 

### 8.3. Trajectories of Fear Recognition Accuracy

To address the hypothesis that CU traits would predict significant change in fear recognition (as indexed by increased accuracy on an emotion recognition task) in the intervention group over and above ICP, LMM was conducted. The LMM analysis used accuracy of fear recognition of the FACES task in the third block of each wave (i.e., free gaze condition), from Wave 1 to Wave 3 as the repeated, dependent measure. In order to examine fear recognition accuracy at the first wave and across the data collection waves, an unconditional growth curve model (Model A) was estimated with no covariates. Following that analysis, study group was added to the model (Model B) to investigate whether group membership predicted fear recognition at Wave 1 and across waves of data collection. A final model (Model C), including time, group, ICP, CU traits, gender, relevant two-way interactions between the predictors, and one three-way interaction between group, time, and CU traits was tested to investigate the predictive utility of these variables, especially of CU traits controlling for the other variables.

Fixed effects and fit comparison statistics for all fear recognition models are reported in Table 4. The unconditional growth model (Model A) indicated that there was a significant time effect for fear recognition across the waves of data collection (*p* < 0.01), including an overall increase in accuracy scores across the waves of data collection. When study group was added to the model (Model B), fit comparison analyses indicated that the inclusion of this predictor resulted in a model that was substantially equivalent with Model A. Although this model predicted FACES accuracy scores at Wave 1 (*p* < 0.001), neither time nor group predicted linear growth in FACES accuracy scores across the waves of data collection. However, when additional hypothesized predictors were added to the model (Model C), fit comparison statistics indicated that the predictors improved the model, with a very strong difference from Model A (see Table 4 for change in AIC and BIC values). Furthermore, with Model C, ICP (*p* < 0.05) and CU traits *p* ≤ 0.05) significantly predicted higher fear recognition accuracy, and a significant two-way interaction between group and CU traits was observed (*p* ≤ 0.05). No other main effects or interactions, including the three-way interaction between group, time, and CU traits were statistically significant.

Given the significant two-way interaction between group and CU traits, this interaction was probed using procedures outlined by Preacher et al. [68]. An online calculator (http://quantpsy.org/interact/hlm2.htm, accessed on 12 March 2011) was used in conjunction with R version 2.14.0 (R Development Core Team, 2011). Examinations of simple intercept analyses indicated that Wave 1 fear recognition scores were significantly different from zero for both study groups. See Table 5 for model coefficients. The interaction simple slopes for Wave 1 are depicted graphically in Figure 6. These findings suggested that fear recognition varied as a function of CU traits and that the magnitude of this relation depended on whether the participant was in the control group or intervention group at baseline. Specifically, individuals in the intervention group with higher levels of CU traits showed the highest fear recognition at the third block of Wave 1. This difference in the intervention group and control group can be explained, in part, by the negative association between CU traits and Wave 1 FACES fear recognition for the control group and the positive association between CU traits and Wave 1 fear recognition for the intervention group. Moreover, that the three-way interaction between group, CU traits, and time was not significant indicated that differences between study groups observed at Wave 1 did not vary across the three waves of data collection. Thus, the higher level of fear recognition observed in the intervention group at Wave 1 was stable across the waves of data collection. Taken together, the hypothesis that CU traits would confer a unique vulnerability for fear recognition over and above ICP was supported. As predicted, fear recognition scores improved the most for participants in the intervention group with high levels of CU traits. That is, these individuals showed the most benefit from the intervention.

### 8.4. CU Traits and Trajectories of Attention to Emotional Cues 

LMM analyses were also used to examine the hypotheses that CU traits and group membership would predict increased attention to non-facial distress cues and non-facial threat cues, as indexed by increased facilitation scores relative to Wave 1. Against predictions, none of the variables in any of the models predicted either Wave 1 facilitation to distress or change in the variable. Similar results were observed for facilitation to threat models. 

### 8.5. CU Traits and Trajectories of Empathy

Fixed effects LMM analyses were also used to examine the remaining hypothesis, that CU traits would predict increased empathic responding, as indexed by improvements in the PT, EC, and PD subscales of the IRI. None of these models showed significant change across waves. Thus, the hypothesis that CU traits would predict increased empathic responding within the intervention group was not supported.

## 9. Discussion

An intervention consisting of instruction to look at the eyes of fearful facial expressions has been successful at temporarily improving the fear blindness characteristic of children with psychopathic traits [2] and patients with bilateral amygdala damage [13]. Missing from this line of research has been investigation of the sustainability of the gains made and an evaluation of the effects of these improvements on other domains. The current study was designed to begin addressing these gaps by examining a pilot test of trajectories of fear recognition accuracy, attention to non-facial distress stimuli, attention to non-facial threat stimuli, and self-reported empathic responding across the span of approximately one week, following the administration of repeated eye-gaze instruction. Of additional interest was whether varying the amount of eye-gaze instruction would have differential effects on the prediction of outcome variables.

Findings from the current study indicated that for youths with elevated levels of CU traits, receiving eye-gaze instruction on more than one occasion contributed to maintaining an increase in fear recognition that was greater than the fear recognition observed in participants receiving the instruction only once. Stability of improvements in fear blindness in the intervention group is consistent with and extends a growing body of research indicating that eye-gaze instruction temporarily corrects the fear blindness evident in individuals with psychopathic traits [2]. However, the current findings also indicated that when the eye-gaze instruction was given repeatedly, improvements in fear recognition were enduring. 

Additionally, we found participants in the intervention group improved in their fear recognition accuracy scores when they had a higher levels of CU traits; this was also true when controlling for ICP and other relevant variables. Improvements in fear recognition for participants in the intervention group with elevated levels of CU traits were greater than the improvements for participants in the control group with elevated levels. The difference in these improvements was sustained across waves of data collection, suggesting that additional eye-gaze instruction sustained the improvements in fear recognition. Consequently, the hypothesis that CU traits would predict significant change in fear recognition in the intervention group, over and above ICP was supported. The results are consistent with the idea that, at least over a short span of time, individuals with the greatest risk for fear blindness maintained improvements in the accuracy for reading fear in other people’s faces after receiving an intensive intervention. This result suggests it may be possible to remediate fear reading deficits as one step in treating emotion deficits in youth high in psychopathic traits. Placement in the intervention group resulted in twice as much deficit-targeting eye-gaze instruction as placement in the control group. This intervention appears to have resulted in the most improvement for those highest in CU traits. 

It is important to note that at Wave 1, the intervention did not have a significant effect on fear recognition in the control group. Given that both groups received the intervention at this wave, one would expect equal responding at the first wave, with group differences occurring in subsequent waves. This finding may be an indication that randomization failed, or that there were additional, unmeasured confounding variables responsible for this group difference. In addition, because of documented gender differences in empathy, distress, and emotionality [59], analyses were conducted both with and without the girls in the sample to see whether gender may account for any effects. Reanalyzing the data without girls did not change the results. Finally, because of the small sample size and concerns about statistical power, we repeated the primary analysis in a simpler form. A linear regression testing the same effects also showed a significant interaction of CU traits by Group in fear recognition at Wave 3 (controlling for earlier Waves, ICP, and Gender), further supporting the main findings.

## 10. Conclusions

The current study aimed to show that impairment of reflexive eye gaze which is aided by the amygdala underlies the fear recognition deficit associated with CU traits in youths. In addition, we aimed to show that an intervention to direct eye gaze to the eye region of faces would lead to better emotion recognition for youths high on CU traits. The two studies were conducted in the UK and the US with youths who were placed in alternative education or in institutions based on their antisocial behavior. Across these two studies, CU traits were associated with a fear recognition deficit. Reflexive gaze to the eye region of faces in youths was lower in those with impulsivity/conduct problems and high on callous traits. Most importantly, Experiment 2 showed that an extended intervention that directed youths to look at the eyes when completing the emotion recognition task worked to ameliorate the fear recognition deficit of those with CU traits, even when accounting for impulsivity/conduct problems. Thus, training youths to fixate on the eyes improved fear recognition and this was sustained for those with CU traits.

People with CU traits show a distinct lack of empathy compared to control samples [4,69]. Additionally, research has shown that people with CU traits (measured as psychopathy) show reduced autonomic responses to stimuli associated with the distress of another individual [70,71], further evidencing their inability to empathize with the emotional states of others. Similarly, Hare [72] found that people with psychopathic tendencies showed lower fear responses in anticipation of an aversive stimulus than controls. Similarly deficient autonomic activity has been shown for children high on CU traits, signifying a ‘fearlessness’ in these children [73]. Linking these emotional experiences to empathy, people with high levels of CU traits may not understand or recognize fearful facial expressions in other people, because of a lack of vicarious emotional experience [74]. In other words, it can be argued that the fear recognition deficit, or fear blindness, demonstrated in children with CU traits is due to an inability to experience fear in the same way as others—attenuating their ability to empathize with others’ fear.

The results of both studies should be considered in light of certain limitations. Primarily, Experiment 2 was a pilot study, which resulted in limited sample size and statistical power. Thus, these results should be considered promising and not definitive. However, our results are consistent with those of previous work, e.g., [2], and represent a first step in the documentation of maintaining increased accuracy in fear recognition, particularly among a difficult to recruit juvenile justice sample, rather than community sample. Thus, further work would benefit from the recruitment of larger samples and additional measures of fear recognition to strengthen and generalize findings.

Despite durability beyond the intervention, similar to other work in this area, changes are documented for only a relatively short timeframe. For example, a parent-child intervention targeting reciprocated eye-gaze showed notable effects in increasing eye-gaze in children high in CU traits during the intervention [75]. This effect, however, was not sustained beyond the treatment phase. Thus, it is important for further efforts at translational strategies target factors that may help to generalize beyond such short-term effects. Some strategies that have evidence for generalizing effects are booster sessions, incorporation of contextual factors and cues that may promote generalization, such as vicarious learning [76]. Additionally, a better understanding of all mechanisms involved in both reflexive gaze and connections to emotion recognition and processing will aid in developing strategies to increase durability of effects. 

Additionally, only the self-report version of the ICU [22] was used, where additional reports (e.g., the parent and/or teacher report ICU) can complement assessment. However, the self-report ICU and the parent report are similarly associated with delinquency [77]. Further studies would benefit from the incorporation of multiple informants. 

The importance of the current study’s findings lies in their confirmation of previous findings as well as in their new contributions to the field. As demonstrated by a previous study [2], the finding that the deficits in reading fear in others’ faces could be reversed suggests that the deficits do not inherently impair knowledge of what fear looks like. Individuals with psychopathic traits can recognize fear in others, though they do not tend to do so spontaneously, without motivation, or without eye-gaze instruction. Most importantly, the observed gains made in fear recognition in Experiment 2 lasted beyond the administration of the study intervention, lending support for their sustainability. Although the intervention did not appear to have transfer effects on attention to non-facial distress stimuli, threat stimuli or empathy, the findings from the current study may be a catalyst for additional research that pairs repeated eye-gaze instruction with additional components that effectively shape the byproducts and correlates of fear blindness deficits. For example, youths with a combination of CU traits and ICP show less autonomic reactivity to empathy-inducing film clips than normal controls [78]. Meffert and colleagues [79] found that specifically instructing individuals with psychopathic traits to try and ‘feel what an actor was feeling’ increased their ability to do so. Additional interventions should target overcoming these processes, so that an empathy-battery may be developed to improve emotion processing tools across the empathy spectrum.

## Figures and Tables

**Figure 1 brainsci-11-01342-f001:**
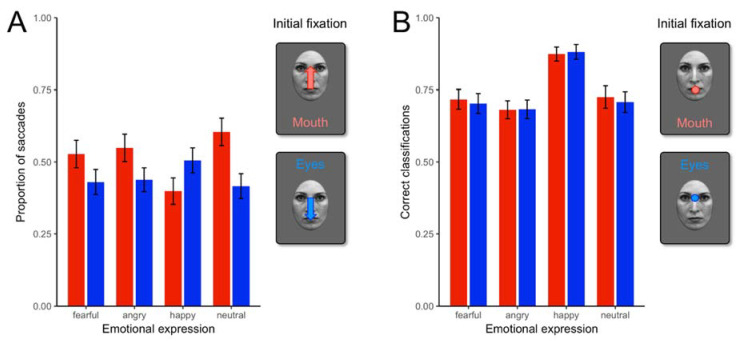
Proportion of saccades as well as the proportion of correct responses across the cohort.

**Figure 2 brainsci-11-01342-f002:**
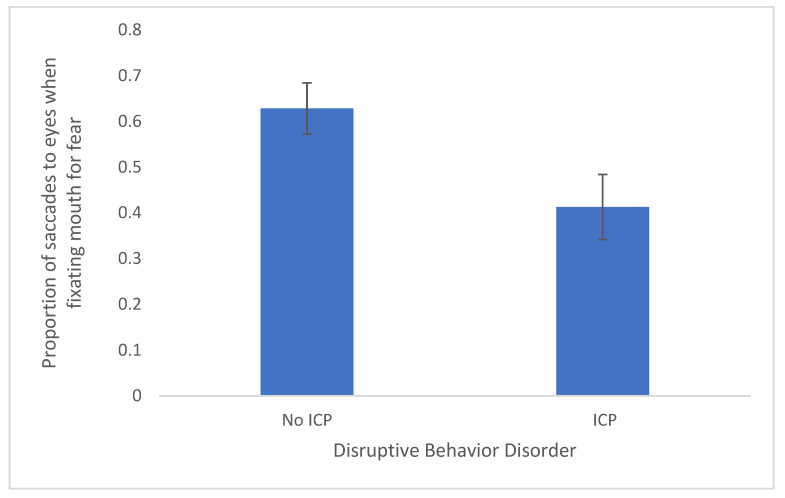
Gaze shifts to the eyes of fearful faces when the mouth was initially fixated by ICP grouping.

**Figure 3 brainsci-11-01342-f003:**
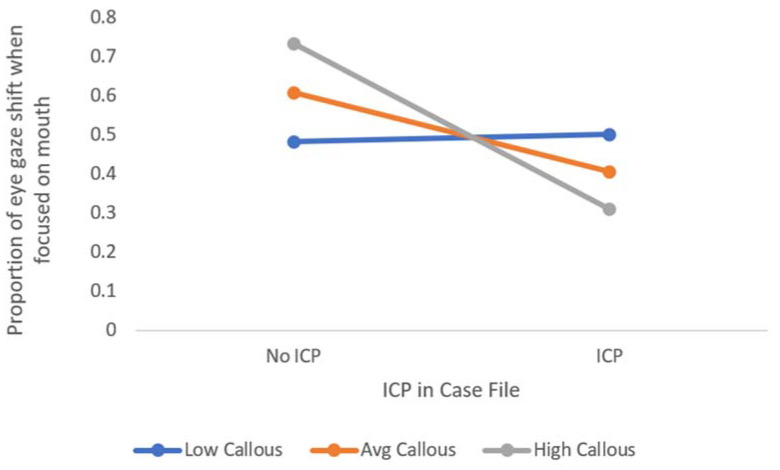
Post-hoc illustration of interaction between the callous facet on the Inventory of Callous Unemotional Traits and ICP in explaining eye-gaze shifts to the eyes for fearful faces.

**Figure 4 brainsci-11-01342-f004:**
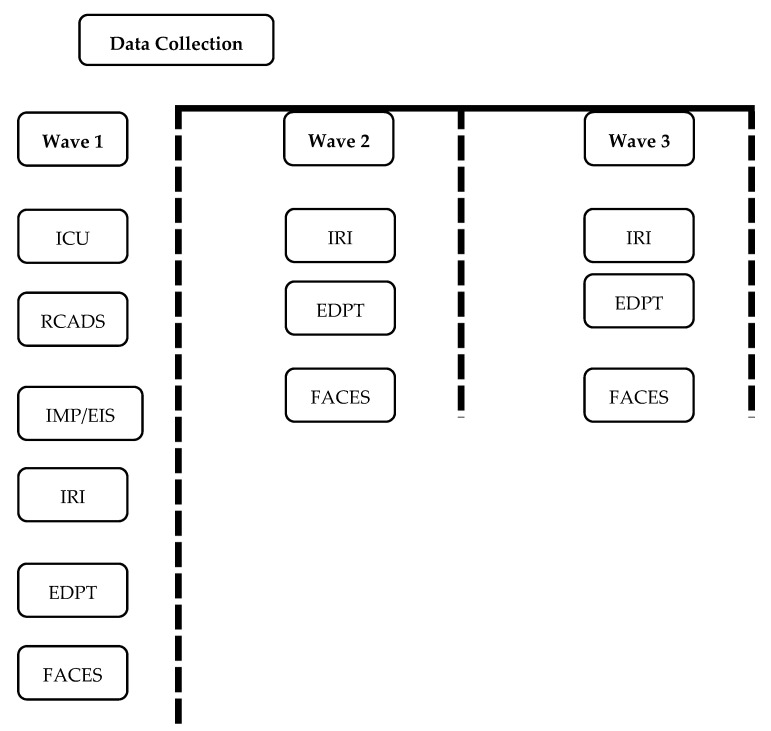
Summary of Waves of Data Collection. Note: ICU = Inventory of Callous-Unemotional Traits; RCADS = Revised Child Anxiety and Depression Scale; IMP = Impulsivity/Conduct Problems Scale of the APSD; EIS = Eysenck Impulsiveness Scale; IRI = Interpersonal Reactivity Index; EDPT = Emotional Pictures Dot-Probe Task; FACES = The University of New South Wales Facial Emotion Task.

**Figure 5 brainsci-11-01342-f005:**
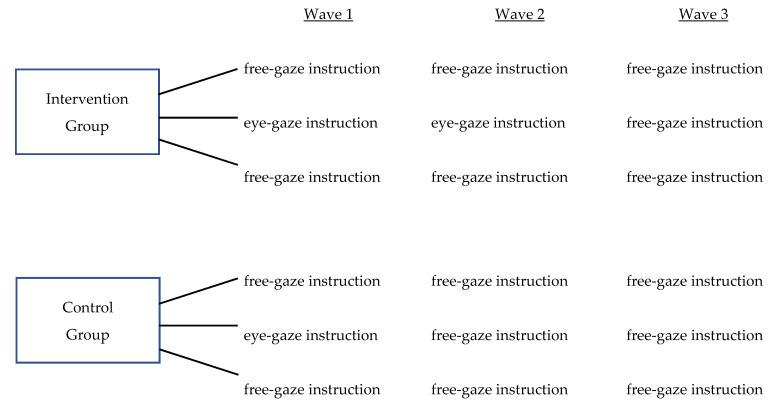
Summary of blocks of FACES task. Note: FACES = The University of New South Wales (UNSW) Facial Emotion Task.

**Figure 6 brainsci-11-01342-f006:**
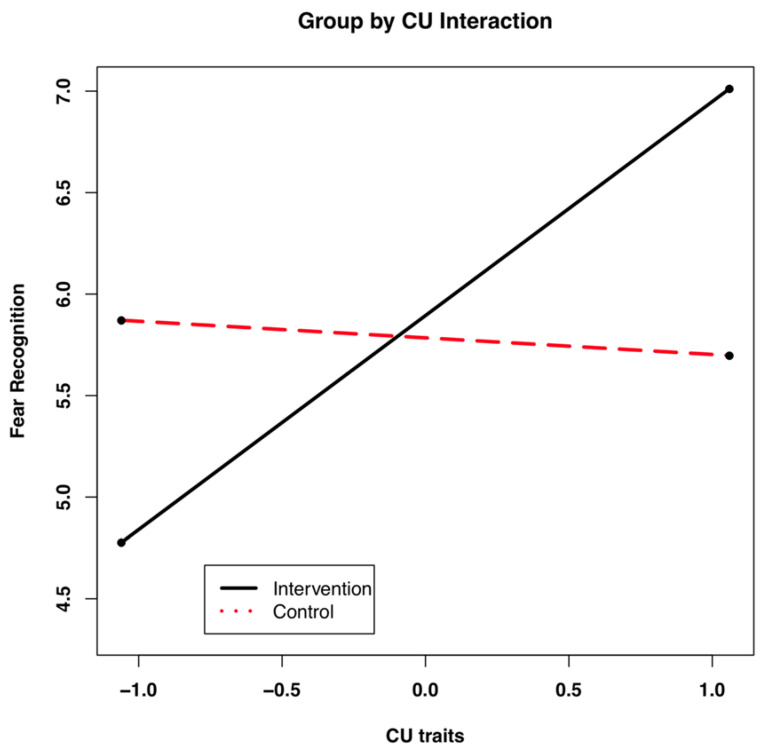
Wave 1 Group by CU Traits Interaction Plot.

**Table 1 brainsci-11-01342-t001:** Correlations (*Kendall’s tau*) among CU traits, anxiety, and age and proportion of reflexive saccades to the eye region when the mouth is fixated (“mouth”) and proportion of reflexive saccades to the mouth region when the eyes are fixated (“eye”).

	1		2		3		4		5		6		7		8	9	10
1. Anger-fix eye	—																
2. Anger-fix mouth	−0.29	**	—														
3. Fear-fix eye	0.63	***	−0.25	**	—												
4. Fear-fix mouth	−0.29	**	0.65	***	−0.27	**	—										
5. Happy-fix eye	0.56	***	−0.12		0.56	***	−0.15		—								
6. Happy-fix mouth	−0.32	***	0.66	***	−0.28	**	0.58	***	−0.18		—						
7. Neutral-fix eye	0.54	***	−0.33	***	0.61	***	−0.35	***	0.54	***	−0.33	***	—				
8. Neutral-fix mouth	−0.33	***	0.65	***	−0.25	**	0.62	***	−0.18		0.51	***	−0.32	***	—		
9. ICU total	0.004		−0.08		−0.09		−0.08		−0.13		−0.01		0.02		−0.06	—	
10. Anxiety	−0.18		0.11		−0.10		0.10		−0.06		0.13		0.02		0.03	0.05	—
11. Age	−0.03		0.09		−0.07		−0.05		0.05		0.07		−0.11		−0.08	0.09	−0.14

** *p* < 0.01, *** *p* < 0.001.

**Table 2 brainsci-11-01342-t002:** Correlations (*Kendall’s tau*) among CU traits, anxiety, and age and emotion recognition when eyes were fixated and when the mouth was fixated.

	1		2		3		4		5	6	7		8	9	10
1. Anger-fix eye	—														
2. Anger-fix mouth	0.47	***	—												
3. Fear-fix eye	0.26	**	0.26	**	—										
4. Fear-fix mouth	0.24	*	0.14		0.47	***	—								
5. Happy-fix eye	0.10		0.02		0.23	*	0.15		—						
6. Happy-fix mouth	0.26	*	0.16		0.15		0.10		0.16	—					
7. Neutral-fix eye	0.28	**	0.35	***	0.43	***	0.2	*	0.17	0.09	—				
8. Neutral-fix mouth	0.18		0.08		0.31	**	0.33	***	0.15	−0.06	0.43	***	—		
9. ICU total	0.02		0.08		−0.13		−0.20	*	−0.16	−0.07	−0.03		−0.18	—	
10. Anxiety	0.04		−0.04		0.01		0.11		−0.03	−0.09	0.03		0.06	0.05	—
11. Age	−0.01		−0.06		−0.12		0.082		0.19	0.27	−0.11		−0.10	0.09	−0.14

* *p* < 0.05, ** *p* < 0.01, *** *p* < 0.001.

**Table 3 brainsci-11-01342-t003:** Means, Standard Deviations, and Correlations among Primary Study Variables.

	CU Traits	Imp (APSD)	Imp (EIS)	W1 FACES Fear	W2 FACES Fear	W3 FACES Fear	W1 Fac Threat	W2 Fac Threat	W3 Fac Threat	IRI PT	IRI EC	M (SD)
CU Traits	--											27.65 (9.84)
Imp (APSD)	0.51 **	--										6.77 (2.06)
Imp (EIS)	0.24	0.61 **	--									15.47 (2.92)
W1 FACES Fear	−0.16	−0.03	−0.01	--								5.30 (0.99)
W2 FACES Fear	0.22	0.39 *	0.17	0.63 **	--							5.73 (0.69)
W3 FACES Fear	0.14	0.33	0.10	0.73 **	0.90 **	--						5.83 (0.65)
W1 Fac Threat	0.06	0.05	0.06	0.00	0.00	−0.01	--					17.62 (89.23)
W2 Fac Threat	0.07	0.21	−0.04	0.62 **	0.77 **	0.76 **	−0.04	--				−4.00 (107.91)
W3 Fac Threat	−0.04	−0.27	−0.14	−0.31	−0.55 **	−0.55 **	0.04	−0.41 *	--			−9.38 (133.57)
IRI PT	−0.45 *	−0.46 *	−0.23	0.32	0.04	0.12	0.29	0.18	0.14	--		12.83 (4.92)
IRI EC	−0.56 **	−0.43 *	−0.20	−0.02	−0.15	−0.10	0.11	0.00	0.11	0.62 **	--	15.60 (6.20)

* *p* < 0.05, ** *p* < 0.01, CU = callous unemotional, Imp = impulsivity, Fac = Facilitation, IRI = Interpersonal Reactivity Index, PT = Perspective taking, EC = Empathic concern.

**Table 4 brainsci-11-01342-t004:** Fixed Effects and Comparison Statistics for Models Predicting FACES Fear Accuracy.

Fixed Effects	Model A	Model B	Model C
Intercept	5.56 ***	5.56 ***	5.98 ***
Time	0.13 **	0.01	0.12
Group		−0.00	−0.37
CU Traits			2.64 *
ICP: APSD			0.25 *
Impulsivity: EIS			−0.12
Sex			0.38
Group × Time		0.08	−0.02
Group × CU Traits			−1.51 *
CU Traits × Time			−0.29
Group × Time × CU Traits			0.19
**Model Comparisons**			
∆ deviance (AIC) from Model A		−2.90	31.23
∆ deviance (AIC) from Model B			34.13
∆ deviance (BIC) from Model A		−2.76	33.74
∆ deviance (BIC) from Model B			36.50

Note: * *p* ≤ 0.05, ** *p* < 0.01, *** *p* < 0.001.

**Table 5 brainsci-11-01342-t005:** Simple Intercepts and Slopes for Interactive Model Predicting Intervention Group’s FACES Fear Accuracy.

Parameters	Estimate	S.E.	z-Score
**Simple Intercepts:**			
Intervention Group	5.72	0.23	24.87 ***
Control Group	1.22	0.56	2.19 *
**Simple Slopes:**			
Intervention Group	5.80	0.24	24.54 ***
Control Group	−0.10	0.42	0.22 +

+ *p* < 0.10, * *p* < 0.05, *** *p* < 0.001.

## Data Availability

Study data are available on request from the LCMC and TRS.

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
