# Peer review of "Reflexive Gaze Shifts and Fear Recognition Deficits in Children with Callous-Unemotional Traits and Impulsivity/Conduct Problems"

_brainsci, 2021, doi:10.3390/brainsci11101342_

Round 1

Reviewer 1 Report

The authors investigate reflexive gaze shifts and fear recognition deficits in children with callous-unemotional traits and impulsivity/conduct problems. Overall, this manuscript is of good quality. I only have a few minor comments.

  1. It would be nice if the authors could discuss related neural circuit mechanisms. For example, the loss of inhibitory control (GABAergic interneurons as well as neurons the RMTg) to midbrain dopamine neurons is heavily involved in punishment learning and impulsivity.
  2. One reference for fear processing is missing. JE Hassell Jr et al., Neuroscience letters, 2019

Author Response

Thanks to the reviewer for the positive feedback and thoughtful comments.

Point 1 - It would be nice if the authors could discuss related neural circuit mechanisms. For example, the loss of inhibitory control (GABAergic interneurons as well as neurons the RMTg) to midbrain dopamine neurons is heavily involved in punishment learning and impulsivity.

We agree that the introduction would be improved by including further discussion on the neural circuit mechanisms. We have added to the following paragraph in section 1.1 to discuss some of the research in this area. 

"People with CU traits may show deficits in reading fearful expressions in faces, simply because they have reduced shifting of attention (via eye gaze) towards the eye region of faces [2]. The eye regions of faces contain the most salient cues for fear recognition – there is greater proportion of the visibility of the sclera of the eyes which acts as an aid [9]. Using eye-tracking methodology, impaired recognition of fear was associated with reduced sampling of the eye region of other people [10]. Recognition of fear and other facial cues is achieved via a distributed neural system e.g., [11]. For example, as noted by Jhou et al. [12], several related systems are involved including visual (middle/superior and medial temporal gyrus), prefrontal (medial frontal gyrus and middle frontal gyrus), and limbic (amygdala, parahippocampal gyrus). The work of these authors also shows core activation in those areas in response to facial stimuli. Additionally, low inhibitory control is, in part, related to the hippocampal system via GABAergic interneurons and rostromedial tegmental nucleus in its role as a “brake” in the midbrain dopamine system [12]. Thus, there are both focal areas of interest and distributed systems that are involved in processing and that may be related to fear blindness. Certainly, if fearful faces are construed as important to learning to inhibit one’s behaviour that is leading to others being afraid, then the involvement of these systems in inhibition and punishment reactivity is key to our understanding. Interestingly, the behavioral findings in children with CU traits parallel observations from patients with bilateral amygdala damage, who show reduced gaze orienting towards the eye region of other people, including the sclera of the eyes [13], [14] as well as deficits in the recognition of fearful facial expressions [15]. On this basis, it has been speculated that the amygdala may be dysfunctional in individuals with CU traits [16], [17]."

Point 2 - One reference for fear processing is missing. JE Hassell Jr et al., Neuroscience letters, 2019

Regarding the suggestion to add JE Hassell Jr et al., Neuroscience letters, 2019, we are unsure where this would fit in our current narrative. We are happy to include it, but it seems to relate to the perception of threat (to self) rather than distress/fear. As such, we were unsure how the reviewer thought that OCT might be linked to the perception of fear in others, although it does seem to be linked to ADHD in one paper. Some clarification would be helpful.

Reviewer 2 Report

The authors represent a creative collaboration of child psychopathologists and neuroscientists with expertise if reflexive gaze shifting and the quality of the research reflects this.

I only have one point for the authors.  The results have important implications for reflexive gaze shifting and emotion recognition to be targeted in innovative interventions. However, the authors stop at the level of experimental studies in which these behaviours are targeted in short sessions using computer stimuli eg programming attention to the eyes can perturb emotion recognition. However, recent attempts to move these experimental manipulations into real world interventions that produce sustained change have so far produced mixed results. As an example, my own work has recently shown we can produce immediate changes in eye gaze with parents but it was not sustained beyond the treatment phase. Other researchers have had some success. I think the paper would benefit from some recognition of these growing attempts as we move toward translational projects.

Otherwise, very nice work.   

Author Response

We express our thanks to the reviewer for the helpful comments and positive feedback for our team!

Point 1 - I only have one point for the authors.  The results have important implications for reflexive gaze shifting and emotion recognition to be targeted in innovative interventions. However, the authors stop at the level of experimental studies in which these behaviours are targeted in short sessions using computer stimuli eg programming attention to the eyes can perturb emotion recognition. However, recent attempts to move these experimental manipulations into real world interventions that produce sustained change have so far produced mixed results. As an example, my own work has recently shown we can produce immediate changes in eye gaze with parents but it was not sustained beyond the treatment phase. Other researchers have had some success. I think the paper would benefit from some recognition of these growing attempts as we move toward translational projects.

We agree that we need to recognize where research has gained traction but that attempts to move forward toward translational strategies are needed. We have added the following paragraph to Section 9. Conclusions:

Despite durability beyond the intervention, similar to other work in this area, changes are documented for only a relatively short timeframe. For example, a parent-child intervention targeting reciprocated eye-gaze showed notable effects in increasing eye-gaze in children high in CU traits during the intervention [75]. This effect, however, was not sustained beyond the treatment phase. Thus, it is important for further efforts at translational strategies target factors that may help to generalize beyond such short-term effects. Some strategies that have evidence for generalizing effects are booster sessions, incorporation of contextual factors and cues that may promote generalization, such as vicarious learning [76]. Additionally, a better understanding of all mechanisms involved in both reflexive gaze and connections to emotion recognition and processing will aid in developing strategies to increase durability of effects.

[75]        M. R. Dadds, T. English, S. Wimalaweera, O. Schollar-Root, and D. J. Hawes, “Can reciprocated parent–child eye gaze and emotional engagement enhance treatment for children with conduct problems and callous-unemotional traits: a proof-of-concept trial,” J. Child Psychol. Psychiatry, vol. 60, no. 6, pp. 676–685, 2019, doi: 10.1111/jcpp.13023.

[76]        J. L. Kleberg, I. Selbing, D. Lundqvist, B. Hofvander, and A. Olsson, “Spontaneous eye movements and trait empathy predict vicarious learning of fear,” Int. J. Psychophysiol. Off. J. Int. Organ. Psychophysiol., vol. 98, no. 3 Pt 2, pp. 577–583, Dec. 2015, doi: 10.1016/j.ijpsycho.2015.04.001.